# Racial and ethnic differences in epigenetic aging: The National Health and Nutrition Examination Survey, 1999–2002

Belinda L. Needham[1]*, Nicole Gladish[2], Hanyang Shen[2], Yongmei Liu[3], Jennifer A. Smith[1,4], Bhramar Mukherjee[5,6], Xiang Zhou[5], David H. Rehkopf[7,8,9,10]

1 Department of Epidemiology, University of Michigan, Ann Arbor, Michigan, United States of America, 2 Department of Epidemiology and Population Health, Stanford University, Stanford, California, United States of America, 3 Department of Medicine, Division of Cardiology, Duke University, Durham, North Carolina, United States of America, 4 Survey Research Center, Institute for Social Research, University of Michigan, Ann Arbor, Michigan, United States of America, 5 Department of Biostatistics, University of Michigan, Ann Arbor, Michigan, United States of America, 6 Department of Biostatistics, Yale University, New Haven, Connecticut, United States of America, 7 Department of Health Policy, Stanford University School of Medicine, Stanford, California, United States of America, 8 Department of Medicine (Primary Care and Population Health), Stanford University, Stanford, California, United States of America, 9 Department of Pediatrics, Stanford University, Stanford, California, United States of America, 10 Department of Sociology, Stanford University, Stanford, California, United States of America

* needhamb@umich.edu

## Abstract

### Background

Accelerated biological aging due to differences in socially patterned exposures has been proposed as a mechanism underlying racial and ethnic disparities in morbidity and mortality. Research exploring this hypothesis has been limited by a lack of consensus regarding the measurement of biological aging.

### Objective

The goal of this study is to examine self-reported race and ethnicity as a predictor of 13 measures of epigenetic aging.

### Methods

Data are from the National Health and Nutrition Examination Survey (1999–2002), a nationally representative study of US residents aged two months and older. The analytic sample includes 2,402 adults aged 50–84 with epigenetic data. The exposure is self-reported race and ethnicity, and the outcomes are 13 measures of epigenetic aging trained on different aging phenotypes.

### Results

In linear regression models controlling for age, age-squared, gender, and nativity, White respondents had higher epigenetic aging than Black respondents (the

**Data availability statement:** The data underlying the results presented in the study are available from the National Health and Nutrition Examination website (https://wwwn.cdc.gov/nchs/nhanes/dnam).

**Funding:** This research was supported by the National Institute on Minority Health and Health Disparities (https://www.nimhd.nih.gov/, R01MD011721, MPI: Needham & Rehkopf). The funder did not play any role in the study design, data collection and analysis, decision to publish, or preparation of the manuscript.

**Competing interests:** The authors have declared that no competing interests exist.

reference group) for six out of seven measures trained on chronological age (Hannum: b = 1.98, 95% CI = 1.43, 2.54; Horvath: b = 0.75, 95% CI = 0.09, 1.40; Weidner: b = 1.15, 95% CI = 0.30, 2.01; Vidal-Bralo: b = 2.30, 95% CI = 1.76, 2.84; SkinBlood: b = 0.85, 95% CI = 0.28, 1.43; Zhang: b = 0.58, 95% CI = 0.40, 0.76) and for one measure trained on telomere length (b = −0.17, 95% CI = −0.20, −0.14). In contrast, White respondents had lower epigenetic aging than Black respondents for three out of four measures trained on physiological age (GrimAge: b = −1.33, 95% CI = −2.01, −0.64; DunedinPoAm: b = −0.03, 95% CI = −0.04, −0.01; GrimAge2: b = −1.97, 95% CI = −2.74, −1.20) and for one measure trained on stem cell divisions (b = −0.01, 95% CI = −0.01, −0.01). Fewer differences in epigenetic aging were observed when comparing Mexican American, other Hispanic, and another race or ethnicity respondents to Black respondents.

## Conclusions

White respondents had higher epigenetic aging than Black respondents for measures trained on chronological age, whereas the opposite was true for measures trained on physiological age. More work is needed to validate measures of epigenetic aging in non-White populations and to determine whether these measures are associated with health-related outcomes similarly across racial and ethnic groups.

## Introduction

Racial and ethnic disparities in morbidity and mortality are well established [1], but the molecular mechanisms have not been fully elucidated [2,3]. According to the weathering hypothesis, Black Americans are at increased risk for age-related diseases and premature death due to accelerated biological aging, which is caused by cumulative exposure to material hardship and psychosocial stressors [4,5]. To date, however, a lack of consensus regarding the measurement of biological aging has limited empirical analysis of the weathering hypothesis [6–8]. In this study, we use data from the National Health and Nutrition Examination Survey to examine self-reported race and ethnicity–a social construct that reflects an individual's position within the racialized social hierarchy of the United States [9]–as a predictor of epigenetic aging among a nationally representative sample of non-Hispanic Black (Black), non-Hispanic White (White), Mexican-American, other Hispanic, and another race or ethnicity adults aged 50–84.

Epigenetic alterations can regulate genetic expression without altering the underlying DNA sequence and are considered a hallmark of aging [10,11]. Over the past decade, researchers have created a variety of epigenetic aging measures based on epigenome-wide patterns of DNA methylation (DNAm) associated with different aging phenotypes [11]. DNAm is the most commonly studied epigenetic mechanism, involving the covalent transfer of a methyl group to a C-5 position of the cytosine ring of DNA [12]. One group of epigenetic aging measures, which we refer to as *DNAm*

*chronological age measures*, were trained to predict chronological age. Often described as epigenetic clocks, examples of measures in this group include the Hannum [13] and Horvath [14] clocks. A second group, which we refer to as *DNAm physiological age measures*, were trained to predict a variety of physiological and behavioral measures, such as frailty biomarkers and smoking. Examples of measures in this group include GrimAge [15] and DunedinPoAm [16]. A third group, which we refer to as *DNAm biomarker of aging measures*, were trained to predict a specific biomarker of aging. Examples include the Yang measure, which was trained to predict stem cell divisions [17], and the Telomere measure, which was trained to predict leukocyte telomere length [18].

Epigenetic aging measures are associated with a wide range of health outcomes that disproportionately affect Black Americans, including obesity [19,20], diabetes [21], dyslipidemia [22], cardiovascular disease [23], cognitive decline [24], and mortality [25]. Previous research suggests that measures of epigenetic aging that were trained on markers of physiological age are better predictors of common disease outcomes than measures that were trained on chronological age [26]. In addition to associations with health outcomes, measures of epigenetic aging are also associated with numerous environmental, psychosocial, and behavioral risk factors, such as air pollution [27], neighborhood poverty [28], socioeconomic disadvantage [29], early life adversity [30], discrimination [31], and physical inactivity [32], that Black Americans are more likely to be exposed to as a result of racism [33,34]. Therefore, epigenetic aging may provide a link between racially patterned health risk exposures and health outcomes. It is important to note, however, that measures of epigenetic aging were created in exclusively or predominantly White samples, which may limit generalizability to non-White populations [6].

The first goal of this study is to examine self-reported race and ethnicity as a predictor of 13 measures of epigenetic aging among adults aged 50–84. Drawing on the weathering hypothesis [4,5], we expect to find that Black respondents will have increased epigenetic aging relative to White, Mexican American, other Hispanic, and another race or ethnicity respondents for all measures except Telomere, given previous evidence of longer telomere length among Black Americans [35]. The second goal of this study is to determine the extent to which racial and ethnic differences in epigenetic aging are attenuated after adjusting for social, economic, and behavioral factors that are hypothesized to be on the causal pathway from the social subordination of Black Americans to accelerated biological aging.

## Methods

### Data and sample

Data are from the National Health and Nutrition Examination Survey (NHANES), 1999–2002. Launched in 1960, NHANES is a repeated cross-sectional study conducted by the National Center for Health Statistics at the Centers for Disease Control and Prevention (CDC) to provide national estimates of the health and nutritional status of the US civilian non-institutionalized population. NHANES 1999–2002 includes a nationally representative sample of 21,004 individuals aged two months and older. Respondents were selected using a four-stage sampling design: (1) primary sampling units (PSUs) consisting primarily of single counties, (2) area segments within PSUs, (3) households within segment areas, and (4) individuals within households. An average of 2–3 individuals per household were sampled. Low-income individuals, adults aged 60 and over, Black Americans, and Mexican Americans were oversampled.

All NHANES 1999–2002 participants aged 20 and over (n = 10,291) were asked to provide DNA samples to establish a national probability sample of genetic material for future research. Seventy-six percent of eligible participants provided viable DNA and consented to future genetic research. A subset of these participants was selected for epigenetic analysis, including a random sample of approximately one-half of all White participants aged 50 and over and all Black, Mexican American, other Hispanic, and another race or ethnicity participants aged 50 and over. The analytic sample includes 516 Black, 943 White, 706 Mexican American, 154 other Hispanic, and 83 another race or ethnicity respondents aged 50–84 with non-missing data on the epigenetic aging measures (total n = 2,402). Respondents over age 84 were excluded because age is top-coded at 85. Human subjects approval was provided by the Institutional Review Board at the CDC and the University of Michigan.

## DNA methylation

Aliquots of purified DNA were provided by the Division of Health and Nutrition Examination Surveys at the National Center for Health Statistics. DNA was extracted from whole blood using standard procedures, and samples were stored at −80 C. The DNAm assay was performed in the Duke Molecular Genomics Core at Duke University. Bisulfite conversion of DNA was carried out using manufacturer's recommendations. Briefly, 500ng of purified DNA was bisulfite treated using a Zymo EZ DNA Methylation kit (cat# D5001, Zymo Research, Irvine, CA, USA) using PCR conditions for Illumina's Infinium Methylation assay (95°C for 30 seconds, 50°C for 60minutes x16 cycles). Data were produced on the Illumina Infinium MethylationEPIC BeadChips (cat# WG317–1001, Illumina, San Diego, CA, USA). A total of 4 μL of bisulfite converted DNA was hybridized to Illumina beadchips using manufacturer's protocols. The samples were denatured and amplified overnight for 20–24 hours. Fragmentation, precipitation, and resuspension of the samples followed overnight incubation, prior to hybridization to EPIC BeadChips for 16–24 hours. BeadChips were then washed to remove any unhybridized DNA and labelled with nucleotides to extend the primers to the DNA sample. Following the Infinium HD Methylation protocol, the BeadChips were imaged using the Illumina iScan system (Illumina, San Diego, CA, USA). DNA samples were coded, and the lab was blinded to all other measurements in the study. The CDC conducted a quality control review prior to linking the DNAm data to the NHANES 1999–2002 public use data files.

## Epigenetic aging

Thirteen measures of epigenetic aging are available in NHANES, including seven DNAm chronological age measures (Hannum [13], Horvath [14], Weidner [36], Vidal-Bralo [37], Lin [38], SkinBlood [39], and Zhang [40]), four DNAm physiological age measures (PhenoAge [41], GrimAge [15], DunedinPoAm [16], and GrimAge2 [42]), and two DNAm biomarkers of aging measures (Yang [17] and Telomere [18]). Given the limited body of research on racial and ethnic differences in epigenetic aging, we chose to examine all available measures. A detailed description of the procedures used to construct these measures is available on the NHANES website (https://wwwn.cdc.gov/nchs/nhanes/dnam/).

## Race and ethnicity

Race and Hispanic origin were self-reported and recoded by NHANES staff into five categories: non-Hispanic Black, non-Hispanic White, Mexican American, other Hispanic, and another race or ethnicity, including multi-racial. In this study, self-reported race and ethnicity is conceptualized as a social construct that reflects an individual's position within the racialized social hierarchy of the United States [9]. Black respondents serve as the reference category because the weathering hypothesis predicts that biological aging will be most pronounced in this group.

## Demographic characteristics

Demographic characteristics include chronological age (in years), gender (female, male), and nativity (foreign born, US born).

## Social and economic characteristics

Social and economic characteristics include marital status (never married, separated, divorced, or widowed; married or living with a partner), educational attainment (less than high school, high school, some college, college degree), poverty income ratio (PIR–calculated as the ratio of income to the poverty threshold for a household of a given size and composition; < 1, 1–1.99, 2–4.99, 5+), and longest held occupation (white collar and professional; white collar, semi-routine; blue collar, high skill; blue collar, semi-routine; never worked).

## Health behaviors

Health behaviors include smoking (never smoker, former smoker less than 30 pack years, former smoker 30–59 pack years, former smoker 60 + pack years, current smoker less than 30 pack years, current smoker 30–59 pack years, current

smoker 60 + pack years), alcohol consumption (abstainer, moderate drinker [1–2 drinks/day for a woman or 1–3 drinks/day for a man], heavy drinker [>2 drinks/day for a woman or >3 drinks/day for a man]), physical activity (inactive [no reported moderate or vigorous activity for ≥10 minutes in past 30 days], active), and diet (2015 Healthy Eating Index [HEI] [43] quintiles).

## Analysis plan

First, we performed multiple imputation in the full NHANES 1999–2002 sample using nonparametric random forest to account for missing data on study covariates (missing data ranged from 0% for race and ethnicity, age, and gender to 11.62% for PIR) [44]. Next, we calculated descriptive statistics for all study variables and examined correlations between chronological age and each of the measures of epigenetic aging, as well as correlations among each of the epigenetic aging measures. We then conducted a series of linear regression models to examine race and ethnicity as a predictor of epigenetic aging and to determine the extent to which racial and ethnic differences in epigenetic aging are attenuated after adjusting for social, economic, and behavioral factors that may be on the causal pathway from race and ethnicity to epigenetic aging. Following Krieger et al. [45], we analyzed raw epigenetic age as the outcome controlling for chronological age as a covariate and accounting for potential non-linearity in the association between chronological age and epigenetic age by inclusion of a squared term for chronological age. This approach is equivalent to analyzing measures of epigenetic age acceleration that are constructed by regressing epigenetic age on chronological age and saving the residuals [45]. We considered several covariate adjustment models. In Model 1, we regressed epigenetic age on the race and ethnicity categories, controlling for chronological age, age-squared, gender, and nativity. In Model 2, we added controls for marital status, educational attainment, PIR, and occupation. In Model 3, we added controls for smoking, alcohol consumption, physical activity, and diet. In sensitivity analyses, we performed a complete case analysis and adjusted for blood cell composition (% lymphocytes, % monocytes, % segmented neutrophils, % eosinophils, and % basophils), which was measured directly in NHANES. Models accounted for the complex sampling design of NHANES by incorporating strata and PSU indicators, as well as sample weights constructed for the epigenetic subsample [46]. This ensures that estimates are representative of the US population aged 50–84. Analyses were conducted in RStudio (version 2024.04.2 + 764) using the R programming language (version 4.4.1).

## Results

Unweighted descriptive statistics for demographic characteristics, social and economic characteristics, and health behaviors are shown in Table 1. The sample is 21.5% Black, 39.3% White, 29.4% Mexican American, 6.4% other Hispanic, and 3.5% another race or ethnicity. Mean age is 65.1 years, and 48.8% of the sample is female. Three-quarters of respondents were born in the US. Over 60% of respondents are married or living with a partner, 45.6% have less than a high school education, and 15.6% live below the federal poverty threshold. A majority of respondents reported blue collar occupations. Over half of respondents are current or former smokers, 57.0% are moderate or heavy drinkers, 51.2% are physically inactive, and 35.4% are in the lowest diet quality quintile (HEI quintiles were calculated in the full NHANES 1999–2002 sample).

Table 2 presents unweighted descriptive statistics for the epigenetic aging measures, as well as Pearson correlations with chronological age. Among the DNAm chronological age measures, mean years of epigenetic age range from 53.8 (Weidner) to 66.4 (Zhang), and correlations with chronological age range from 0.56 (Weidner) to 0.87 (Zhang). Among the DNAm physiological age measures that calculate epigenetic age in years, means range from 54.8 (PhenoAge) to 71.5 (GrimAge2), and correlations with chronological age range from 0.76 (PhenoAge) to 0.82 (GrimAge). For DunedinPoAm, the mean pace of aging is 1.11, and the correlation with chronological age is 0.06. Among the DNAm biomarkers of aging measures, the mean pcgtAge for Yang is 0.1, and the mean kilobase pairs

**Table 1. Unweighted descriptive statistics for demographic characteristics, social and economic characteristics, and health behaviors, NHANES 1999-2002 (n = 2,402).**

| Measure | Mean (SE) or n (%) |
|---|---|
| *Demographic Characteristics* | |
| Race and ethnicity | |
| Non-Hispanic Black | 516 (21.5%) |
| Non-Hispanic White | 943 (39.3%) |
| Mexican American | 706 (29.4%) |
| Other Hispanic | 154 (6.4%) |
| Another race or ethnicity | 83 (3.5%) |
| Age (years) | 65.1 (9.3) |
| Gender | |
| Female | 1172 (48.8%) |
| Male | 1230 (51.2%) |
| Nativity | |
| US born | 1794 (74.7%) |
| Foreign born | 608 (25.3%) |
| *Social and Economic Characteristics* | |
| Marital status | |
| Never married, separated, divorced, or widowed | 846 (35.2%) |
| Married or living with a partner | 1556 (64.8%) |
| Educational attainment | |
| Less than high school | 1096 (45.6%) |
| High school | 496 (20.6%) |
| Some college | 440 (18.3%) |
| College degree or higher | 370 (15.4%) |
| Poverty income ratio | |
| <1 | 374 (15.6%) |
| 1-1.90 | 715 (29.8%) |
| 2-4.99 | 914 (38.1%) |
| 5+ | 399 (16.6%) |
| Longest held occupation | |
| White collar and professional | 537 (22.4%) |
| White collar, semi-routine | 410 (17.1%) |
| Blue collar, high skill | 387 (16.1%) |
| Blue collar, semi-routine | 1005 (41.8%) |
| Never worked | 63 (2.6%) |
| *Health Behaviors* | |
| Smoking | |
| Never smoker | 1094 (45.5%) |
| Former smoker less than 30 pack years | 591 (24.6%) |
| Former smoker 30–59 pack years | 229 (9.5%) |
| Former smoker 60 + pack years | 103 (4.3%) |
| Current smoker less than 30 pack years | 184 (7.7%) |
| Current smoker 30–59 pack years | 153 (6.4%) |
| Current smoker 60 + pack years | 48 (2.0%) |

*(Continued)*

**Table 1.** (Continued)

| Measure | Mean (SE) or n (%) |
|---|---|
| Alcohol consumption | |
| Abstainer | 1033 (43.0%) |
| Moderate drinker | 1283 (53.4%) |
| Heavy drinker | 86 (3.6%) |
| Physical activity | |
| Inactive | 1231 (51.2%) |
| Active | 1171 (48.8%) |
| Healthy Eating Index (HEI)[a] | |
| 1st quintile | 851 (35.4%) |
| 2nd quintile | 128 (5.3%) |
| 3rd quintile | 120 (5.0%) |
| 4th quintile | 212 (8.8%) |
| 5th quintile | 1091 (45.4%) |

[a]HEI quintiles calculated in the full NHANES 1999–2002 sample.

for Telomere is 6.6. The correlation with chronological age is 0.23 for Yang and −0.59 for Telomere. Correlations between the epigenetic aging measures are shown in Table 3 and range from −0.00 (Weidner and DunedinPoAm) to 0.99 (GrimAge and GrimAge2).

Results of the linear regression models examining racial and ethnic differences in 13 measures of epigenetic aging, controlling for chronological age, age-squared, gender, and nativity are shown in Model 1 of Table 4. White respondents had higher epigenetic aging than Black respondents for six out of seven DNAm chronological age measures (Hannum: b = 1.98, 95% CI = 1.43, 2.54; Horvath: b = 0.75, 95% CI = 0.09, 1.40; Weidner: b = 1.15, 95% CI = 0.30, 2.01; Vidal-Bralo: b = 2.30, 95% CI = 1.76, 2.84; SkinBlood: b = 0.85, 95% CI = 0.28, 1.43; Zhang: b = 0.58, 95% CI = 0.40, 0.76), while Mexican American respondents had higher epigenetic aging than Black respondents for four out of seven DNAm chronological age measures (Hannum: b = 3.50, 95% CI = 2.30, 4.69; Vidal-Bralo: b = 1.26, 95% CI = 0.54, 1.97; SkinBlood: b = 1.31, 95% CI = 0.30, 2.32; Zhang: b = 0.54, 95% CI = 0.18, 0.90). In contrast, White respondents had lower epigenetic aging than Black respondents for three out of four DNAm physiological age measures (GrimAge: b = −1.33, 95% CI = −2.01, −0.64; DunedinPoAm: b = −0.03, 95% CI = −0.04, −0.01; GrimAge2: b = −1.97, 95% CI = −2.74, −1.20), while Mexican American respondents had higher epigenetic aging than Black respondents for one measure (PhenoAge: b = 2.52, 95% CI = 1.22, 3.82). Among the DNAm biomarkers of aging measures, White and other Hispanic respondents had lower epigenetic aging than Black respondents on the Yang measure (White: b = −0.01, 95% CI = −0.01, −0.01; other Hispanic: b = −0.01, 95% CI = −0.01, −0.00), while all racial and ethnic groups had higher epigenetic aging (i.e., shorter telomere length) than Black respondents on the Telomere measure (White: b = −0.17, 95% CI = −0.20, −0.14; Mexican American: b = −0.19, 95% CI = −0.24, −0.14; other Hispanic: b = −0.16, 95% CI = −0.22, −0.10; another race or ethnicity: b = −0.22, 95% CI = −0.27, −0.17). As shown in Table 4, associations were similar after adding controls for marital status, educational attainment, PIR, and occupation in Model 2 and after adding controls for smoking, alcohol consumption, physical activity, and diet in Model 3. Findings from the complete case sensitivity analysis were similar to the main findings. Results of the sensitivity analysis adjusting for measured blood cell composition are shown in Table S1.

**Table 2. Unweighted descriptive statistics and Pearson correlations with age for epigenetic aging measures, NHANES 1999–2002 (n = 2,402).**

| Measure | Training Phenotype | Training Sample Size | Training Age Range | Training Race and Ethnicity Groups | Tissue(s) Examined | Unit | Mean | Min, Max | Correlation with Age |
|---|---|---|---|---|---|---|---|---|---|
| *DNAm Chronological Age Measures* | | | | | | | | | |
| Hannum | Chronological age | 656 | 19-101 years | 65% White; 35% Hispanic | Whole blood | Years | 66.2 | 25.1, 105.4 | 0.81 |
| Horvath | Chronological age | 3,931 | 0-100 years | 68% White; 17% Unknown; 6% Hispanic; 6% Black; 3% Asian | 51 tissue types | Years | 66.0 | 25.5, 102.8 | 0.79 |
| Weidner | Chronological age | 446 | 0-79 years | 100% White | Whole blood | Years | 53.8 | 28.1, 123.7 | 0.56 |
| Vidal-Bralo | Chronological age | 390 | 20-79 years | 100% White | Whole blood | Years | 59.8 | 32.5, 98.6 | 0.62 |
| Lin | Chronological age | 575 | 0-79 years | 100% White | Whole blood | Years | 56.5 | 8.1, 120.2 | 0.75 |
| Skin-Blood | Chronological age | 896 | −0.28-94 years | 45% White; 37% Unknown; 10% Indigenous; 7% Black; 1% Hispanic | Whole and cord blood, epidermal and ex vivo tissues | Years | 63.5 | 16.9, 91.7 | 0.86 |
| Zhang | Chronological age | 13,661 | 2.2-104 years | 100% White | Blood and saliva | Years | 66.4 | 47.5, 78.9 | 0.87 |
| *DNAm Physiological Age Measures* | | | | | | | | | |
| Pheno-Age | Phenotypic age (10 biomarkers) | 456 | 21-91 years | 100% White | Whole blood | Years | 54.8 | 5.1, 100.04 | 0.76 |
| GrimAge | Chronological age, smoking, DNAm biomarkers of proteins associated with mortality | 1,731 | 59-73 years | 100% White | Whole blood | Years | 65.6 | 43.5, 92.4 | 0.82 |
| Dunedin-PoAm | Pace of aging (18 biomarkers) | 810 | 38 years | 93% White; 7% Māori | Whole blood | Pace of aging | 1.11 | 0.8, 1.5 | 0.06 |
| Grim-Age2 | Chronological age, smoking, DNAm biomarkers of proteins associated with mortality | 1,833 | 59-73 years | 100% White | Whole blood | Years | 71.5 | 48.5, 97.5 | 0.78 |
| *DNAm Biomarkers of Aging Measures* | | | | | | | | | |
| Yang | Stem cell divisions | 656 | 19-101 years | 65% White; 35% Hispanic | Whole blood | pcgtAge | 0.1 | 0.0, 0.2 | 0.23 |
| Telomere | Leukocyte telomere length | 2,256 | 22-93 years | 81% Black; 19% White | Leukocytes | Kilobase pairs | 6.6 | 5.3, 7.8 | −0.59 |

**Table 3. Pearson correlation matrix for epigenetic aging measures, NHANES 1999-2002 (n = 2,402).**

| | Hannum | Horvath | Weidner | Vidal-Bralo | Lin | SkinBlood | Zhang | PhenoAge | GrimAge | Dunedin PoAm | GrimAge2 | Yang | Telomere |
|---|---|---|---|---|---|---|---|---|---|---|---|---|---|
| DNAm Chronological Age Measures | | | | | | | | | | | | | |
| Hannum | 1 | | | | | | | | | | | | |
| Horvath | 0.87 | 1 | | | | | | | | | | | |
| Weidner | 0.62 | 0.61 | 1 | | | | | | | | | | |
| Vidal-Bralo | 0.71 | 0.73 | 0.70 | 1 | | | | | | | | | |
| Lin | 0.84 | 0.85 | 0.64 | 0.72 | 1 | | | | | | | | |
| SkinBlood | 0.92 | 0.89 | 0.61 | 0.71 | 0.84 | 1 | | | | | | | |
| Zhang | 0.92 | 0.90 | 0.61 | 0.70 | 0.85 | 0.96 | 1 | | | | | | |
| DNA Physiological Age Measures | | | | | | | | | | | | | |
| PhenoAge | 0.84 | 0.84 | 0.57 | 0.73 | 0.80 | 0.84 | 0.84 | 1 | | | | | |
| GrimAge | 0.74 | 0.72 | 0.46 | 0.62 | 0.67 | 0.74 | 0.75 | 0.75 | 1 | | | | |
| DenedinPoAm | 0.17 | 0.13 | −0.00 | 0.20 | 0.15 | 0.09 | 0.10 | 0.28 | 0.44 | 1 | | | |
| GrimAge2 | 0.72 | 0.68 | 0.43 | 0.60 | 0.64 | 0.72 | 0.72 | 0.76 | 0.99 | 0.49 | 1 | | |
| DNAm Biomarkers of Aging Measures | | | | | | | | | | | | | |
| Yang | 0.33 | 0.29 | 0.28 | 0.05 | 0.19 | 0.30 | 0.27 | 0.19 | 0.16 | −0.20 | 0.15 | 1 | |
| Telomere | −0.73 | −0.63 | −0.50 | −0.56 | −0.60 | −0.66 | −0.69 | −0.65 | −0.63 | −0.19 | −0.60 | −0.33 | 1 |

## Discussion

The weathering hypothesis states that Black Americans experience accelerated biological aging relative to other racial and ethnic groups because of cumulative exposure to material hardship and psychosocial stressors, and that differences in biological aging may explain why Black Americans are at increased risk of age-related diseases and premature mortality [4,5]. While the theoretical construct of weathering has been operationalized in a variety of ways, there is currently no consensus on the best way to measure biological aging [6–8]. Over the past decade, measures of epigenetic aging, which are based on patterns of DNAm across the genome, have emerged as potentially useful empirical indicators of weathering. In this study, we used nationally representative data from NHANES to examine self-reported race and ethnicity as a predictor of 13 measures of epigenetic aging. We also examined the extent to which racial and ethnic differences in epigenetic aging were attenuated after adjusting for social, economic, and behavioral factors that are hypothesized to be on the causal pathway from the social subordination of Black Americans to accelerated biological aging.

We found mixed support for the hypothesis that Black Americans experience accelerated epigenetic aging relative to other racial and ethnic groups. Contrary to expectations, we found that White respondents had higher epigenetic aging than Black respondents for six out of seven measures trained on chronological age. These results are consistent with previous research in the Health and Retirement Study (HRS), which found that White respondents had higher epigenetic aging than Black respondents on the Hannum, Horvath, Weidner, Vidal-Bralo, Lin, and Zhang measures [47]. SkinBlood was not examined in HRS. Also contrary to expectations, we found that Mexican American respondents had higher epigenetic aging than Black respondents on four of the seven DNAm chronological age measures. Because HRS used White respondents as the reference group and we used Black respondents as the reference group, we were unable to compare results for differences between Black and Hispanic respondents across the two studies.

In contrast to the results for the DNAm chronological age measures, we found that White respondents had lower epigenetic aging than Black respondents for three out of four epigenetic aging measures trained on physiological age, which previous research suggests are better predictors of common disease outcomes than measures trained on chronological age [26]. These results were consistent with our hypothesis and similar to findings from HRS, which also includes a

**Table 4. Linear regression models examining racial and ethnic differences in epigenetic aging, NHANES 1999-2002 (n = 2,402).**

| | Model 1 | Model 2 | Model 3 |
|---|---|---|---|
| | Beta (95% CI) | Beta (95% CI) | Beta (95% CI) |
| *DNAm Chronological Age Measures* | | | |
| Hannum | | | |
| Non-Hispanic Black | Ref. | Ref. | Ref. |
| Non-Hispanic White | **1.98 (1.43, 2.54)** | **2.13 (1.47, 2.78)** | **2.09 (1.37, 2.82)** |
| Mexican American | **3.50 (2.30, 4.69)** | **3.17 (1.96, 4.38)** | **3.22 (2.02, 4.42)** |
| Other Hispanic | **2.19 (1.16, 3.23)** | **1.90 (0.82, 2.98)** | **1.90 (0.92, 2.89)** |
| Other race or ethnicity | **2.28 (0.67, 3.88)** | **2.11 (0.49, 3.73)** | **2.01 (0.38, 3.65)** |
| Horvath | | | |
| Non-Hispanic Black | Ref. | Ref. | Ref. |
| Non-Hispanic White | **0.75 (0.09, 1.40)** | **0.77 (0.03, 1.51)** | 0.79 (−0.02, 1.60) |
| Mexican American | 0.64 (−0.39, 1.66) | 0.47 (−0.63, 1.57) | 0.46 (−0.63, 1.55) |
| Other Hispanic | −0.04 (−0.98, 0.91) | −0.04 (−1.11, 1.02) | −0.10 (−1.07, 0.87) |
| Other race or ethnicity | 0.43 (−0.95, 1.81) | 0.35 (−1.14, 1.83) | 0.23 (−1.26, 1.73) |
| Weidner | | | |
| Non-Hispanic Black | Ref. | Ref. | Ref. |
| Non-Hispanic White | **1.15 (0.30, 2.01)** | **1.03 (0.04, 2.01)** | 0.99 (−0.14, 2.13) |
| Mexican American | 0.82 (−0.19, 1.82) | 0.72 (−0.42, 1.85) | 0.67 (−0.44, 1.77) |
| Other Hispanic | −1.15 (−3.07, 0.77) | −1.24 (−3.16, 0.68) | −1.35 (−3.33, 0.64) |
| Other race or ethnicity | −1.81 (−4.44, 0.81) | −1.99 (−4.79, 0.80) | −1.92 (−4.78, 0.94) |
| Vidal-Bralo | | | |
| Non-Hispanic Black | Ref. | Ref. | Ref. |
| Non-Hispanic White | **2.30 (1.76, 2.84)** | **2.26 (1.64, 2.87)** | **2.22 (1.52, 2.92)** |
| Mexican American | **1.26 (0.54, 1.97)** | **1.07 (0.33, 1.80)** | **1.10 (0.39, 1.82)** |
| Other Hispanic | 0.83 (−0.22, 1.87) | 0.69 (−0.40, 1.77) | 0.67 (−0.37, 1.71) |
| Another race or ethnicity | −0.36 (−1.49, 0.77) | −0.53 (−1.67, 0.61) | −0.45 (−1.60, 0.71) |
| Lin | | | |
| Non-Hispanic Black | Ref. | Ref. | Ref. |
| Non-Hispanic White | 0.99 (−0.01, 2.00) | 0.89 (−0.13, 1.91) | 0.93 (−0.13, 1.98) |
| Mexican American | 0.76 (−0.69, 2.21) | 0.56 (−1.00, 2.11) | 0.45 (−1.12, 2.01) |
| Other Hispanic | −0.22 (−2.32, 1.87) | −0.11 (−2.26, 2.04) | −0.28 (−2.35, 1.79) |
| Other race or ethnicity | −0.73 (−2.56, 1.11) | −0.79 (−2.64, 1.06) | −0.94 (−2.71, 0.82) |
| SkinBlood | | | |
| Non-Hispanic Black | Ref. | Ref. | Ref. |
| Non-Hispanic White | **0.85 (0.28, 1.43)** | **0.88 (0.27, 1.50)** | **0.95 (0.25, 1.64)** |
| Mexican American | **1.31 (0.30, 2.32)** | 0.99 (−0.02, 2.00) | **1.03 (0.02, 2.04)** |
| Other Hispanic | **1.00 (0.08, 1.92)** | 0.86 (−0.14, 1.86) | 0.82 (−0.06, 1.69) |
| Other race or ethnicity | 1.28 (−0.08, 2.64) | 1.16 (−0.27, 2.59) | 1.02 (−0.41, 2.44) |
| Zhang | | | |
| Non-Hispanic Black | Ref. | Ref. | Ref. |
| Non-Hispanic White | **0.58 (0.40, 0.76)** | **0.55 (0.34, 0.77)** | **0.56 (0.32, 0.80)** |
| Mexican American | **0.54 (0.18, 0.90)** | **0.44 (0.05, 0.82)** | **0.44 (0.06, 0.82)** |
| Other Hispanic | **0.54 (0.21, 0.87)** | **0.50 (0.13, 0.87)** | **0.47 (0.14, 0.81)** |
| Other race or ethnicity | 0.34 (−0.15, 0.83) | 0.26 (−0.26, 0.77) | 0.20 (−0.30, 0.71) |

*(Continued)*

**Table 4.** (Continued)

| | Model 1 | Model 2 | Model 3 |
|---|---|---|---|
| *DNAm Physiological Age Measures* | | | |
| PhenoAge | | | |
| Non-Hispanic Black | Ref. | Ref. | Ref. |
| Non-Hispanic White | 0.75 (−0.06, 1.56) | **1.23 (0.32, 2.13)** | **1.14 (0.21, 2.06)** |
| Mexican American | **2.52 (1.22, 3.82)** | **2.08 (0.81, 3.35)** | **2.20 (0.98, 3.43)** |
| Other Hispanic | 1.26 (−0.16, 2.68) | 0.81 (−0.66, 2.28) | 0.81 (−0.68, 2.29) |
| Other race or ethnicity | 1.23 (−1.31, 3.76) | 1.15 (−1.17, 3.47) | 1.13 (−1.28, 3.54) |
| GrimAge | | | |
| Non-Hispanic Black | Ref. | Ref. | Ref. |
| Non-Hispanic White | **−1.33 (−2.01, −0.64)** | −0.43 (−1.09, 0.22) | **−0.95 (−1.48, −0.42)** |
| Mexican American | −0.53 (−1.60, 0.53) | −0.69 (−1.50, 0.12) | −0.60 (−1.25, 0.05) |
| Other Hispanic | −0.72 (−1.93, 0.48) | −1.17 (−2.35, 0.00) | **−1.37 (−2.44, −0.29)** |
| Other race or ethnicity | −0.71 (−1.99, 0.58) | −0.32 (−1.46, 0.81) | −0.72 (−1.79, 0.34) |
| DunedinPoAm | | | |
| Non-Hispanic Black | Ref. | Ref. | Ref. |
| Non-Hispanic White | **−0.03 (−0.04, −0.01)** | −0.01 (−0.03, 0.00) | **−0.02 (−0.03, −0.01)** |
| Mexican American | 0.00 (−0.02, 0.02) | 0.00 (−0.02, 0.01) | 0.00 (−0.01, 0.01) |
| Other Hispanic | 0.01 (−0.02, 0.03) | −0.00 (−0.03, 0.02) | −0.00 (−0.02, 0.02) |
| Other race or ethnicity | 0.00 (−0.03, 0.03) | 0.01 (−0.02, 0.04) | 0.01 (−0.02, 0.03) |
| GrimAge2 | | | |
| Non-Hispanic Black | Ref. | Ref. | Ref. |
| Non-Hispanic White | **−1.97 (−2.74, −1.20)** | **−0.91 (−1.61, −0.20)** | **−1.39 (−2.00, −0.79)** |
| Mexican American | −0.34 (−1.50, 0.83) | −0.56 (−1.44, 0.31) | −0.42 (−1.17, 0.33) |
| Other Hispanic | −0.38 (−1.70, 0.93) | −0.94 (−2.21, 0.33) | −1,05 (−2.26, 0.16) |
| Other race or ethnicity | −0.77 (−2.40, 0.85) | −0.32 (−1.66, 1.01) | −0.70 (−2.09, 0.68) |
| *DNAm Biomarkers of Aging Measures* | | | |
| Yang | | | |
| Non-Hispanic Black | Ref. | Ref. | Ref. |
| Non-Hispanic White | **−0.01 (−0.01, −0.01)** | **−0.01 (−0.01, −0.01)** | **−0.01 (−0.01, −0.01)** |
| Mexican American | 0.00 (−0.00, 0.00) | −0.00 (−0.01, 0.00) | −0.00 (−0.01, 0.00) |
| Other Hispanic | **−0.01 (−0.01, −0.00)** | **−0.01 (−0.01, −0.00)** | **−0.01 (−0.01, −0.00)** |
| Other race or ethnicity | 0.00 (−0.01, 0.01) | 0.00 (−0.01, 0.01) | 0.00 (−0.01, 0.01) |
| Telomere | | | |
| Non-Hispanic Black | Ref. | Ref. | Ref. |
| Non-Hispanic White | **−0.17 (−0.20, −0.14)** | **−0.19 (−0.22, −0.16)** | **−0.17 (−0.20, −0.15)** |
| Mexican American | **−0.19 (−0.24, −0.14)** | **−0.18 (−0.23, −0.13)** | **−0.18 (−0.22, −0.13)** |
| Other Hispanic | **−0.16 (−0.22, −0.10)** | **−0.15 (−0.20, −0.09)** | **−0.14 (−0.20, −0.08)** |
| Other race or ethnicity | **−0.22 (−0.27, −0.17)** | **−0.23 (−0.28, −0.18)** | **−0.22 (−0.27, −0.17)** |

Note: Model 1 adjusts for chronological age, age-squared, sex, and nativity. Model 2 adjusts for all covariates in Model 1, plus marital status, educational attainment, poverty income ratio, and occupation. Model 3 adjusts for all covariates in Models 1 and 2, plus smoking, alcohol consumption, physical activity, and diet. Beta coefficients and 95% confidence intervals in bold are significant at the p < 0.05 level.

nationally representative sample of older adults [47]. However, our finding of no significant difference between Black and White respondents on the PhenoAge measure differed from two previous studies, which found that Black respondents had higher PhenoAge acceleration than White respondents [41,48]. Differences in study results may be due to differences in sample characteristics, including gender composition and age range, or study covariates.

Results for the DNAm biomarkers of aging measures were also consistent with expectations. We found that White and other Hispanic respondents had lower epigenetic aging than Black respondents on the Yang measure, which was trained on stem cell divisions. Results from HRS were similar when comparing Black and White respondents [47]. Results for the Telomere measure, which was trained on leukocyte telomere length, suggested that all racial and ethnic groups had lower epigenetic aging than Black respondents. While this finding is inconsistent with the weathering hypothesis, it was expected due to the large body of evidence demonstrating that Black Americans have longer telomeres than White Americans [35].

With a few exceptions, we found that racial and ethnic differences in epigenetic aging were not attenuated after adjusting for marital status, educational attainment, PIR, occupation, smoking, alcohol consumption, physical activity, and diet. These results are consistent with a previous study of Black adults, which found that health risk behaviors did not mediate associations between four measures of adversity (education, income, neighborhood disadvantage, and discrimination) and epigenetic aging as measured by GrimAge [49].

### Strengths, limitations, and directions for future research

A key strength of this large, nationally representative study is that results are generalizable to US adults aged 50–84. While it is possible that data collected in more recent years would produce different results, a strength of NHANES 1999–2002 is that it includes up to 20 years of mortality follow-up data from the National Death Index. In future work, we will be able to explore epigenetic aging measures as potential mediators of racial and ethnic disparities in premature, all-cause, and cause-specific mortality. Another strength is the focus on measures of epigenetic aging as indicators of the theoretical construct of weathering. Epigenetic alterations are considered a hallmark of aging [10], and previous research has established that epigenetic aging measures are associated with a wide range of health risk factors [27–32] and health outcomes [19–25] that disproportionately affect Black Americans.

A limitation of existing epigenetic aging measures is that they were created in exclusively or predominantly White samples, which may limit generalizability to non-White populations [6]. More work is needed to validate these measures in different racial and ethnic groups. Although race and ethnicity are not meaningful biological categories, we acknowledge that self-reported race and ethnicity is correlated with genetic ancestry [50]. We were unable to control for this potential confounder because measures of genetic ancestry are not available in NHANES. Thus, future studies in NHANES should evaluate whether epigenetic aging measures are associated with health-related outcomes similarly across racial and ethnic groups. Finally, because the risk of death increases with age, the focus on adults aged 50+ may have introduced survivor bias, particularly among Black Americans, who experience high levels of premature mortality [51]. To address this limitation, future studies should examine racial and ethnic differences in epigenetic aging among younger samples.

### Conclusions

In a nationally representative sample of older adults, we found that White respondents had higher epigenetic aging than Black respondents for measures trained on chronological age, whereas the opposite was true for most measures trained on physiological age. Previous research suggests that measures of epigenetic aging that were trained on markers of physiological age are better predictors of common disease outcomes than measures that were trained on chronological age and may, therefore, have greater clinical relevance [26]. However, given that the epigenetic aging measures in this study were created in exclusively or predominantly White samples, more work is needed to establish the utility of existing measures for research on racial and ethnic health disparities. In the future, it may be possible to use measures of

epigenetic aging to monitor racial and ethnic disparities in biological aging across the life course and to evaluate the effectiveness of interventions to reduce these disparities [6].

## Supporting information

**S1 Table. Linear Regression Models Examining Racial and Ethnic Differences in Epigenetic Aging with Adjustment for Blood Cell Composition, NHANES 1999–2002 (n = 2,402).** Model 1 adjusts for chronological age, age-squared, gender, nativity, and blood cell composition (% lymphocytes, % monocytes, % segmented neutrophils, % eosinophils, and % basophils). Model 2 adjusts for all covariates in Model 1, plus marital status, educational attainment, poverty income ratio, and occupation. Model 3 adjusts for all covariates in Models 1 and 2, plus smoking, alcohol consumption, physical activity, and diet. Beta coefficients and 95% confidence intervals in bold are significant at the $p < 0.05$ level. (DOCX)

## Acknowledgments

The authors wish to thank Dima Chaar and Lisha Lin for their assistance with the literature review, and Jody McLean and Carolyn Neal for their assistance with preparation of the data set. We would also like to acknowledge the assistance of the Molecular Genomics Core at the Duke Molecular Physiology Institute, Duke University School of Medicine, for the generation of the EPIC array data.

## Author contributions

**Conceptualization:** Belinda L. Needham, David H. Rehkopf.

**Data curation:** Nicole Gladish, Yongmei Liu.

**Formal analysis:** Nicole Gladish, Hanyang Shen.

**Funding acquisition:** Belinda L. Needham, Yongmei Liu, Jennifer A. Smith, Bhramar Mukherjee, Xiang Zhou, David H. Rehkopf.

**Methodology:** Belinda L. Needham, David H. Rehkopf.

**Project administration:** Belinda L. Needham, David H. Rehkopf.

**Resources:** Belinda L. Needham.

**Supervision:** Belinda L. Needham, David H. Rehkopf.

**Writing – original draft:** Belinda L. Needham.

**Writing – review & editing:** Nicole Gladish, Hanyang Shen, Yongmei Liu, Jennifer A. Smith, Bhramar Mukherjee, Xiang Zhou, David H. Rehkopf.

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
