## [Decision Letter · Decision Letter 0]

Dear Dr. Needham,

Thank you for submitting your manuscript to PLOS ONE. After careful consideration, we feel that it has merit but does not fully meet PLOS ONE’s publication criteria as it currently stands. Therefore, we invite you to submit a revised version of the manuscript that addresses the points raised during the review process.

We look forward to receiving your revised manuscript.

Kind regards,

Li Yang, M.D.

Academic Editor

PLOS ONE

**Journal Requirements:**

This research was supported by the National Institute on Minority Health and Health Disparities (https://www.nimhd.nih.gov/, R01MD011721, MPI: Needham & Rehkopf). The funder did not play any role in the study design, data collection and analysis, decision to publish, or preparation of the manuscript.

This research was supported by the National Institute on Minority Health and Health Disparities (R01MD011721, MPI: Needham & Rehkopf). The authors wish to thank Dima Chaar and Lisha Lin for their assistance with the literature review, and Jody McLean and Carolyn Neal for their assistance with preparation of the data set.

This research was supported by the National Institute on Minority Health and Health Disparities (https://www.nimhd.nih.gov/, R01MD011721, MPI: Needham & Rehkopf). The funder did not play any role in the study design, data collection and analysis, decision to publish, or preparation of the manuscript.

4. We notice that your supplementary tables are included in the manuscript file. Please remove them and upload them with the file type 'Supporting Information'. Please ensure that each Supporting Information file has a legend listed in the manuscript after the references list.

Reviewers' comments:

Reviewer's Responses to Questions

**Comments to the Author**

1. Is the manuscript technically sound, and do the data support the conclusions?

Reviewer #1: Yes

Reviewer #2: Yes

Reviewer #3: Yes

2. Has the statistical analysis been performed appropriately and rigorously?

Reviewer #1: Yes

Reviewer #2: Yes

Reviewer #3: Yes

3. Have the authors made all data underlying the findings in their manuscript fully available?

Reviewer #1: Yes

Reviewer #2: No

Reviewer #3: Yes

4. Is the manuscript presented in an intelligible fashion and written in standard English?

Reviewer #1: Yes

Reviewer #2: No

Reviewer #3: Yes

**Reviewer #1:**  The manuscript is technically robust, with a well-designed study using high-quality NHANES data and rigorous methods, including DNA methylation analysis as a marker of epigenetic aging. Appropriate controls, replication, and adjustments for confounders enhance the validity of the findings. The study addresses a critical gap by linking epigenetic markers to racial and ethnic disparities, with a timely focus on the "weathering hypothesis" as a plausible mechanism. Statistical analyses are rigorously applied, with clear methods for handling missing data and robust regression models. Data availability is transparent, meeting PLOS ONE’s policies, and the manuscript is well-written, logically structured, and supported by effective tables and figures.

**Reviewer #2:**  "White respondents had higher epigenetic aging than Black respondents (the reference

group) for six out of seven measures trained on chronological age" and "White respondents had

lower epigenetic aging than Black respondents for three out of four measures trained

on physiological age" your article has excellent data and statistics, however it lacks of practical organization to deliver your findings in a pertinent clinical setting. Please elaborate both sentences above and explain the clinical implications of these findings in the first part of your Discussion session.

Please explain this epigenetic findings, taking into account the variance quantitative genetic equation

Vp = Vg + Vepi + Ve

p=phenotype

g=genotype

epi= epigenetics

e=environment

**Reviewer #3: ** 1. How do the epigenetic age measurement tools mentioned in the study specifically reflect biological ageing? Are there other potential biomarkers that could complement these measures?

2. did the study consider other biological mechanisms that may influence racial and ethnic health disparities, such as genetic factors or environmental exposures?

3. is there sample selection bias due to sample selection for NHANES? How was the general applicability of the results ensured?

4. are there specific factors that have a more significant effect on epigenetic age after adjusting for socioeconomic factors? How do these factors affect health outcomes for different ethnic groups?

**Do you want your identity to be public for this peer review?** For information about this choice, including consent withdrawal, please see our Privacy Policy

Reviewer #1: No

Reviewer #2: **Yes: ** VICTORIA BIRD

Reviewer #3: No

---

## [Author Response · Author response to Decision Letter 1]

2 Apr 2025

From our response to reviewers:

Ref: PONE-D-24-45761

Manuscript Title: Racial and ethnic differences in epigenetic aging: The National Health and Nutrition Examination Survey, 1999-2002

We thank the reviewers for their thoughtful comments, which have strengthened the manuscript. Responses to the reviewers’ comments appear below each point.

Reviewer #1:

The manuscript is technically robust, with a well-designed study using high-quality NHANES data and rigorous methods, including DNA methylation analysis as a marker of epigenetic aging. Appropriate controls, replication, and adjustments for confounders enhance the validity of the findings. The study addresses a critical gap by linking epigenetic markers to racial and ethnic disparities, with a timely focus on the "weathering hypothesis" as a plausible mechanism. Statistical analyses are rigorously applied, with clear methods for handling missing data and robust regression models. Data availability is transparent, meeting PLOS ONE’s policies, and the manuscript is well-written, logically structured, and supported by effective tables and figures.

Introduction

The introduction does an admirable job of emphasizing the importance of the study by focusing on the "weathering" hypothesis, which suggests a potential link between accelerated biological aging and racial and ethnic disparities in the U.S. By grounding the argument in relevant literature, the author provides a solid theoretical framework, highlighting the ways in which psychosocial and biological exposures can influence health outcomes. Furthermore, the study is positioned as addressing a key gap in research, namely the absence of consensus on biological aging measures. This makes its purpose both timely and compelling.

Two clear objectives are outlined: investigating racial and ethnic differences in epigenetic aging and examining the role of socioeconomic and behavioral factors in attenuating these differences. The discussion of DNA methylation and its relevance to health outcomes provides a strong foundation for understanding the study's goals. However, the introduction could be enhanced by providing greater detail about the epigenetic markers used and their connection to health disparities, as well as a stronger emphasis on how the predominance of white-based epigenetic measures may limit broader applicability.

We added the following sentence in the Introduction to provide more information about the connection between the various epigenetic aging measures and health outcomes: “Previous research suggests that measures of epigenetic aging that were trained on markers of physiological age are better predictors of common disease outcomes than measures that were trained on chronological age.[26]” We also added the following sentence in the Introduction to acknowledge the fact that epigenetic aging measures were created in predominantly or exclusively White samples: “It is important to note, however, that measures of epigenetic aging were created in exclusively or predominantly White samples, which may limit generalizability to non-White populations.[6]”

Methodology

The methods section is comprehensive, providing clear descriptions of the data sources, sampling strategies, and analytical approaches. The use of NHANES data, combined with robust protocols for DNA methylation analysis, enhances the study's reliability and replicability. Moreover, the inclusion of minority oversampling and adjustments for demographic and behavioral covariates ensures a thoughtful and inclusive approach to the analysis.

While the technical details are well-documented, some areas could be expanded for clarity. For instance, the rationale behind selecting specific epigenetic measures is not thoroughly explained, leaving readers to infer their relevance. Similarly, the mention of multiple imputations for missing data would benefit from additional context on its validation and potential impact on results. Lastly, a more explicit discussion of potential biases in the epigenetic measures—especially considering their development within predominantly white populations—would strengthen this section.

We added the following statement to the Methods to explain the selection of epigenetic aging measures: “Thirteen measures of epigenetic aging are available in NHANES…Given the limited body of research on racial and ethnic differences in epigenetic aging, we chose to examine all available measures.” We also added a complete case sensitivity analysis, which produced results similar to the main findings. Information about potential biases in the epigenetic measures was added to the Introduction, as noted above. Full details about the racial and ethnic composition of the training samples is available in Table 2.

Results

The results are clearly presented, with detailed statistical summaries and thoughtful integration of tables and figures. The focus on multiple epigenetic aging measures provides a nuanced analysis of racial and ethnic differences. However, certain results that deviate from the initial hypothesis, such as instances where white participants exhibit greater epigenetic aging, could be better contextualized.

Regarding the unexpected finding that White respondents had higher epigenetic aging for six out of seven measures trained on chronological age, it is important to note that previous research has shown weaker associations of these measures with common disease outcomes when compared to measures of epigenetic aging trained on markers of physiological age (which showed results consistent with the weathering hypothesis). To address this point, we added the following text in the Conclusions section: “In a nationally representative sample of older adults, we found that White respondents had higher epigenetic aging than Black respondents for measures trained on chronological age, whereas the opposite was true for most measures trained on physiological age. Previous research suggests that measures of epigenetic aging that were trained on markers of physiological age are better predictors of common disease outcomes than measures that were trained on chronological age and may, therefore, have greater clinical relevance.[26]”

While the statistical significance of findings is well-addressed, the practical implications of some results—particularly in measures like telomeres—could be further emphasized. The authors might also consider discussing potential analytical limitations, such as biases introduced by oversampling minorities or the validation gaps in epigenetic measures for non-white populations.

As noted in the Discussion, future studies evaluating whether the epigenetic aging measures are associated with health outcomes similarly across racial and ethnic groups are needed to determine the practical implications of our findings.

We used sample weights designed specifically for the NHANES epigenetic biomarker dataset to ensure that results are generalizable to the US population aged 50+. Thus, oversampling of minorities did not bias study results. To address potential selection bias, we added the following text in the Discussion: “Finally, because the risk of death increases with age, the focus on adults aged 50+ may introduce survivor bias, particularly among Black Americans, who experience high levels of premature mortality.[51] To address this limitation, future studies should examine racial and ethnic differences in epigenetic aging among younger samples.”

Discussion

The discussion builds effectively on the study’s findings, linking results to the theoretical framework and broader literature. The analysis of discrepancies between expected and observed outcomes, such as the unexpected epigenetic aging patterns in white participants, is particularly commendable. This reflective approach enhances the credibility of the study.

Despite these strengths, the discussion could delve deeper into the social and biological factors behind these unexpected results. Additionally, while the study acknowledges racial health disparities, it stops short of exploring how these findings might influence public health policies or interventions. A more explicit connection between epigenetic measures and tangible health outcomes, such as morbidity or mortality rates, would also add depth and relevance.

To address these points, we revised the Conclusions section to say: “In a nationally representative sample of older adults, we found that White respondents had higher epigenetic aging than Black respondents for measures trained on chronological age, whereas the opposite was true for most measures trained on physiological age. Previous research suggests that measures of epigenetic aging that were trained on markers of physiological age are better predictors of common disease outcomes than measures that were trained on chronological age and may, therefore, have greater clinical relevance.[26] However, given that the epigenetic aging measures in this study were created in exclusively or predominantly White samples, more work is needed to establish the utility of existing measures for research on racial and ethnic health disparities. In the future, it may be possible to use measures of epigenetic aging to monitor racial and ethnic disparities in biological aging across the life course and to evaluate the effectiveness of interventions to reduce these disparities.[6] ”

Conclusion

The conclusion succinctly summarizes the study’s contributions, reaffirming its relevance to public health and epigenetic research. By emphasizing the need for validation of epigenetic measures in diverse populations, it sets a clear direction for future studies. However, the practical applications of the findings—particularly for public health strategies—are not fully explored. A more direct discussion of how this research could inform policies to address racial health disparities would significantly enhance its impact.

As noted above, we revised the Conclusions section to address the potential policy relevance of epigenetic aging measures.

Recommendations

To refine the manuscript and ensure greater engagement:

Connect Findings to Broader Implications: Strengthen the link between unexpected results and theoretical or practical explanations, particularly in the discussion.

Expand the Scope of Policy Relevance: Emphasize how the findings could inform interventions or monitoring of racial health disparities.

Address Methodological Gaps: Provide more justification for the selection of epigenetic measures and a deeper discussion of their limitations in racially diverse populations.

Please see above for responses to specific points that are summarized in these three recommendations.

Reviewer #2:

"White respondents had higher epigenetic aging than Black respondents (the reference

group) for six out of seven measures trained on chronological age" and "White respondents had lower epigenetic aging than Black respondents for three out of four measures trained on physiological age" your article has excellent data and statistics, however it lacks of practical organization to deliver your findings in a pertinent clinical setting. Please elaborate both sentences above and explain the clinical implications of these findings in the first part of your Discussion session.

Previous research has shown that measures of epigenetic aging that were trained on markers of physiological age tend to be better predictors of common disease outcomes than measures that were trained on chronological age.1 In this study, racial and ethnic differences in epigenetic aging measures trained on physiological age were consistent with what we know about racial and ethnic differences in morbidity and mortality (i.e., Black respondents had worse outcomes than White respondents), whereas racial and ethnic differences in epigenetic aging measures trained on chronological age were not (i.e., Black respondents had better outcomes than White respondents). We added the following text in the Discussion to address the clinical implications of our findings: “Previous research suggests that measures of epigenetic aging that were trained on markers of physiological age are better predictors of common disease outcomes than measures that were trained on chronological age and may, therefore, have greater clinical relevance.[26]”

Please explain this epigenetic findings, taking into account the variance quantitative genetic equation

Vp = Vg + Vepi + Ve

p=phenotype

g=genotype

epi= epigenetics

e=environment

Because we did not have access to genotype data, we were unable to directly partition variance into genetic (Vg), epigenetic (Vepi), and environmental (Ve) components. However, we acknowledge that all three factors likely contribute to racial and ethnic differences in epigenetic aging. Our findings showed that disparities in epigenetic aging persisted even after controlling for key environmental exposures, including socioeconomic status and health behavior (see Model 3). The persistence of these disparities suggests that environmental factors alone may not fully explain observed differences, indicating a potential role for genetic influences. While we did not have genotype data to integrate in this study, the literature suggests heritability of epigenetic age acceleration.2–5 For this reason, we included a statement in the Discussion acknowledging that genetic ancestry is a potential unmeasured confounder in our analyses.

Reviewer #3:

1. How do the epigenetic age measurement tools mentioned in the study specifically reflect biological ageing? Are there other potential biomarkers that could complement these measures?

As noted in the Introduction, epigenetic alterations are considered a hallmark of aging. The measures of epigenetic aging included in this study were trained on a variety of aging phenotypes. The first group of measures was trained to predict chronological age; the second group of measures was trained to predict physiological and behavioral measures such as blood pressure, cystatin C, and smoking, that are associated with morbidity and mortality; and the third group of measures was trained to predict other biomarkers of aging, such as telomere length. Higher epigenetic age (as determined by these measures) relative to chronological age is hypothesized to reflect accelerated biological aging. Recent work examining 11 measures of epigenetic aging (9 of which were included in our study) found associations with gene regulation in cellular aging pathways (e.g., metabolism, autophagy, immunity), as well as in vivo evidence for associations with mitochondrial dysfunction and cellular senescence.6

Previous research in NHANES has examined other biomarkers of aging, including directly measured telomere length, allostatic load, and homeostatic dysregulation. Although it is beyond the scope of this paper, future studies could examine correlations between epigenetic aging measures and other biomarkers of aging that are available in NHANES.

2. did the study consider other biological mechanisms that may influence racial and ethnic health disparities, such as genetic factors or environmental exposures?

As noted above in our response to Reviewer 2, we did not have access to genotype data. Therefore, we were not able to account for potential confounding due to genetic ancestry (see an acknowledgement of this limitation in the Discussion). While our models accounted for some important environmental exposures, including socioeconomic status and health behavior, there are many additional environmental exposures, such as air pollution and pesticides, that could explain racial and ethnic disparities in epigenetic aging. Given the wealth of environmental exposure data in NHANES, this is an important direction for future research.

3. is there sample selection bias due to sample selection for NHANES? How was the general applicability of the results ensured?

We added the following text in the Discussion to address potential selection bias: “Finally, because the risk of death increases with age, the focus on adults aged 50+ may introduce survivor bias, particularly among Black Americans, who experience high levels of premature mortality.[51] To addre

---

## [Decision Letter · Decision Letter 1]

Racial and ethnic differences in epigenetic aging: The National Health and Nutrition Examination Survey, 1999-2002

PONE-D-24-45761R1

Dear Dr. Needham,

We’re pleased to inform you that your manuscript has been judged scientifically suitable for publication and will be formally accepted for publication once it meets all outstanding technical requirements.

Kind regards,

Li Yang, M.D.

Academic Editor

PLOS ONE

Additional Editor Comments (optional):

Thanks for the authors' efforts to comprehensively improve your manuscript according to editor's and reviewers' comments. I am pleased to inform you that your paper can be accepted for publication now. Thanks for the chance to assess your work. Additionally, many thanks for all the reviewers' precious inputs.

Reviewers' comments:

Reviewer's Responses to Questions

**Comments to the Author**

Reviewer #1: All comments have been addressed

2. Is the manuscript technically sound, and do the data support the conclusions?

Reviewer #1: Yes

3. Has the statistical analysis been performed appropriately and rigorously?

Reviewer #1: Yes

4. Have the authors made all data underlying the findings in their manuscript fully available?

Reviewer #1: Yes

5. Is the manuscript presented in an intelligible fashion and written in standard English?

Reviewer #1: Yes

Reviewer #1: Thank you for your careful revisions. You've addressed all of the previous comments thoroughly, and the manuscript is now much clearer and more complete. The research is solid, the data support the conclusions well, and the statistical analysis has been carried out appropriately.The data availability is in line with journal requirements, and the overall writing is clear and easy to follow. I have no remaining concerns, and I believe the manuscript is ready for publication as it stands.

**Do you want your identity to be public for this peer review?** For information about this choice, including consent withdrawal, please see our Privacy Policy

Reviewer #1: No

---

## [Editor Report · Acceptance letter]

PONE-D-24-45761R1

PLOS ONE

Dear Dr. Needham,

I'm pleased to inform you that your manuscript has been deemed suitable for publication in PLOS ONE. Congratulations! Your manuscript is now being handed over to our production team.

Kind regards,

on behalf of

Dr. Li Yang

Academic Editor

PLOS ONE